



# Manganese incorporation in living (stained) benthic foraminiferal shells: A bathymetric and in-sediment study in the Gulf of Lions (NW Mediterranean).

Shauna Ní Fhlaithearta[1], Christophe Fontanier[2, 3, 4], Frans Jorissen[4], Aurélia Mouret[4],

Adriana Dueñas-Bohórquez[1], Pierre Anschutz[2], Mattias B. Fricker[5], Detlef Günther[5],

Gert J. de Lange[1], Gert-Jan Reichart[1, 6]

[1] Faculty of Geosciences, Utrecht University, Utrecht, The Netherlands.

[2] EPOC, UMR CNRS 5805, University of Bordeaux, Pessac, France

[3] FORAM, Foraminiferal Study Group, F-29200, Brest, France

[4] Université d'Angers, LPG-BIAF , UMR CNRS 6112, 49045 Angers Cedex, France

[5] Laboratory of Inorganic Chemistry, ETH Zurich, 8093 Zurich, Switzerland.

[6] Royal NIOZ, Texel, The Netherlands

Corresponding author: S. Ní Fhlaithearta, s.ni.fhlaithearta@gmail.com

Keywords: Benthic foraminifera, minor/trace metals, calibration study, Gulf of Lions, Mn/Ca

## Abstract

Manganese geochemistry in deep-sea sediments is known to vary greatly over the first

few centimeters, which overlaps with the in-sediment depth habitats of several benthic

foraminiferal species. Here we investigated manganese incorporation in benthic

foraminiferal shell carbonate across a 6-station depth transect in the Gulf of Lions,

NW Mediterranean to unravel the impacts of foraminiferal ecology and Mn pore

water geochemistry. Over this transect water depth increases from 350 to 1987 m,

while temperature (~13˚C) and salinity (~38.5) remained relatively constant.

Manganese concentrations in the tests of living (Rose Bengal stained) benthic

foraminiferal specimens of *Hoeglundina elegans*, *Melonis barleeanus, Uvigerina

mediterranea, Uvigerina peregrina* were measured using laser ablation inductively

coupled mass spectrometry (laser ablation ICP-MS). Pore water manganese

concentrations show a decrease from shallow to deeper waters, which corresponds to

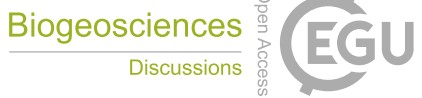
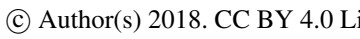


a generally decreasing organic matter flux with water depth. Differences in organic
matter loading at the sediment water interface affects oxygen penetration depth into
the sediment and hence Mn pore water profiles. Mn/Ca values for the investigated
foraminiferal species reflect pore water geochemistry and species-specific
microhabitat in the sediment. The observed degree of variability within a single
species is in–line with know ranges in depth habitat and gradients in redox conditions.
Both Mn/Ca ratio and inter-specific variability hence reflect past Mn cycling and
related early diagenetic processes within the sediment, making this a potential tool for
bottom-water oxygenation and organic-matter fluxes. Dynamics of both in-sediment
foraminiferal depth habitats and Mn cycling, however, limit the application of such a
proxy to settings with relatively stable environmental conditions.

**1. Introduction**
Reconstructing past climate and environmental change largely depends on so-called
proxies. These proxies relate measurable variables in the geological record to target
parameters, such as e.g. temperature, biological productivity and bottom water
oxygenation. The carbonate shells of unicellular protists, foraminifera, are one of the
most utilized signal carriers for reconstructing past environments. Both the census
data of foraminifera and the geochemical composition of the shells are used in this
context. The geochemical composition of the shells is investigated for their stable
isotopic composition as well as for their trace and minor element incorporation. Both
surface and bottom water conditions are reconstructed this way, using planktonic and
benthic foraminiferal species respectively.

Most existing calibrations of trace element uptake in foraminiferal test

carbonate are based on comparing their composition with bottom water conditions




(Elderfield et al., 2006; Nürnberg et al., 1996; Yu and Elderfield, 2007). Many
benthic foraminiferal species live, however, within the sediment and precipitate their
calcium carbonate test in contact with pore water. As a result, the trace metal
composition of pore water exerts a control on the uptake of trace metals in their test.
This effectively links benthic foraminiferal microhabitat preference and pore water
chemistry. On the one hand, this creates complications when using foraminiferal trace
metal ratios for reconstructing bottom water conditions, whereas on the other hand, it
offers the possibility to develop proxies of pore water chemistry in the past.
Linking foraminiferal test chemistry with pore water chemistry requires in-
depth knowledge of, 1) how early diagenesis in sediments affects pore water
chemistry, 2) the habitat preference of the foraminiferal species, 3) foraminiferal
migration (and calcification) within the uppermost sediment layer. In principle, the
chemical composition of living (stained) benthic foraminifera will reflect all these
processes.
For many elements an important inter-specific difference in uptake of trace
metals has been observed (Hintz et al., 2006; Wit et al., 2012), a so-called vital effect.
This implies that in addition to ecology and pore water geochemistry, trace metal
partitioning also needs to be taken into consideration. This requires a comparative
study between locations where all three of these aspects have been quantified.
Reconstructing past pore water trace metal profiles is important since it
provides valuable information on organic carbon degradation and recycling of
nutrients at the seafloor (Van Cappellen and Wang, 1996; De Lange, 1986). Such
diagenetically controlled trace metal profiles are used in quantitative models
constraining oceanic carbon fluxes and burial (Wang and Van Cappellen, 1996).



Knowledge of such profiles in the past could thus help to reconstruct past carbon
cycles.
Benthic foraminiferal species have a specific preference for their depth-habitat
(Jorissen et al., 1995). Some benthic foraminiferal species are limited to a very narrow
environmental in-sediment range, for example, along redox fronts, whereas others
have a wider distribution, thriving under variable conditions and consequently occupy
a broader niche. These differences in depth-habitat preferences could be related to the
presence of different types of metabolism (Koho et al., 2011; Risgaard-Petersen et al.,
2006). As such, trace metal profiles and foraminiferal in-sediment depth habitat can
be related, such as recently proposed in a conceptual (TROXCHEM[3]) model for the
redox sensitive element, manganese, by Koho et al. (2015). Studying the interplay
between benthic foraminiferal habitat preference and incorporation of redox-sensitive
trace elements is key to verifying such models.
Studying manganese bound in foraminiferal shell carbonate lies at the
intersection of foraminiferal ecology and early diagenesis in sediments. Manganese is
a redox sensitive element and exists as Mn- (hydr)oxides in the presence of oxygen.
As oxygen concentrations in the sediment decreases due to ongoing organic matter
remineralization, Mn-(hydr)oxides are reduced to aqueous $Mn^{2+}$, (Froelich et al.,
1979). Manganese in sediments cycles continuously between solid and aqueous state
as a result of upward diffusion of $Mn^{2+}$ and consequent remineralization to Mn-
(hydr)oxides. Hence proxy studies must account for both ecological controls, like
foraminiferal habitat preference, as well as geochemical controls like oxygen
concentrations and organic matter loading (Glock et al., 2012; Groeneveld and
Filipsson, 2013; Koho et al., 2015, 2017; McKay et al., 2015; Reichart et al., 2003).
Notably, both benthic foraminifera and trace metal geochemistry react to organic



matter recycling and bottom water oxygenation (Jorissen et al., 1995). This implies
that locations with contrasting conditions, both low and high bottom-water
oxygenation as well as low and high productivity, are required for testing. Whereas
most of these studies focused on the role of bottom water oxygenation in relatively
oxygen poor settings, here we focus on the well-oxygenated western Mediterranean.

In this study we combine pore water geochemistry, foraminiferal habitat

preference and test geochemistry in an area characterized by well-oxygenated bottom
water conditions and average productivity. Results are compared with earlier studies
from high productivity regimes and low-oxygen conditions at the sediment-water
interface (e.g. Arabian Sea, Koho et al., 2015 and Mediterranean sapropel deposition,
Ní Fhlaithearta et al., 2010). Specifically, we investigate the link between manganese
incorporation and benthic foraminiferal ecology and compare this to the recently
proposed TROXCHEM[3] model (Koho et al., 2015). Four species of living (stained)
foraminifera were sampled along a 6-station bathymetric transect in the Gulf of Lions,
NW Mediterranean. Individuals were picked from a series of in-sediment depths and
analyzed by laser ablation ICP-MS, enabling multiple analyses of single specimens.

**2. Material and methods**

**2.1 Study area and sediment sampling**
Cores were collected with a classical Barnett multicorer (Barnett et al. 1984) at 6
stations in the Gulf of Lions (NW Mediterranean) during the August-September 2006
BEHEMOTH cruise (Fig. 1, Table 1). The 6 stations describe a bathymetric transect
from 350m to 1987m depth. The shallowest site, station F, is bathed in Mediterranean
Intermediate Water (MIW). Stations E (552 m) and D (745 m) are positioned at the





transition of MIW and Western Mediterranean Deep Water (WMDW). Stations C
(980 m), B (1488 m) and A (1987 m) are bathed by the WMDW. Bottom water
temperature is stable through the part of the water column studied here (~13.1°C)
(Xavier Durrieu de Madron, pers. com.). Salinity ranges between 38.4 and 38.5. The
multicorer allowed sampling of the first decimeters of the sediment, the overlying
bottom waters, and an undisturbed sediment-water interface. Cores were sliced for
foraminiferal studies with a 0.5-cm resolution down to 4 cm, followed by 1 cm slices
down to 10 cm depth. Sediments were put in an ethanol-Rose Bengal mixture (95%
ethanol with 1g/l Rose Bengal), in order to identify living (stained) specimens. For
more detailed information about methods, please consult Fontanier et al., (2008).


**2.2 Pore water geochemistry**
Sediment sampling for pore water extraction was carried out under an inert
atmosphere ($N_2$). Hereafter, samples were centrifuged at 3500 rpm for 20 min. The
supernatant was filtered and acidified ($HNO_3$ *s.p.*) for analyzing dissolved metals.
Dissolved Mn concentrations were determined with flame atomic absorption
spectrometry (Perkin Elmer AA 300). Precision for this method is ± 5%. A pore water
subsample was also analyzed for Mn using ICP-MS (Agilent 7500 Series). Relative
precision for this method is 3%. Total alkalinity of pore water was measured at
Utrecht University using an automated titrator (702 SM Titrino, Metrohm) making
Gran plots. Dissolved Inorganic Carbon (DIC) was measured using a Dissolved
Carbon Analyser (Shimadzu, Model TOC-5050A). Carbonate ion concentrations were
calculated using the CO2SYS software (version 01.05; Lewis and Wallace, 1998).



Analytical uncertainty for the alkalinity is about 10 µeq, relative standard deviation
for the DIC analyses is 0.8%.

Oxygen concentration profiles were determined using Clark-type

microelectrodes (Unisense©, Denmark). Labile organic matter was derived from the
sum of lipids, amino acids and sugars measured in the top cm of sediment; for details,
see Fontanier et al., 2008.

**2.3 Foraminiferal trace metal geochemistry**
Foraminiferal trace element concentrations were determined using two laser ablation
ICP-MS systems. Prior to laser ablation, all samples were gently cleaned in methanol
(x1) and UHQ water (x4). Between each rinse, the samples were placed in a sonic
bath for several seconds to thoroughly clean the tests. Benthic foraminifera from 745
m (station D), 980 m (station C), 1488 m (station B) and 1987 m (station A) were
measured at Utrecht University using a deep UV (193nm) ArF excimer laser (Lambda
Physik) with GeoLas 200Q optics. Ablation was performed at a pulse repetition rate
of 10 Hz, and energy density of 1.4 J/cm$^2$, with a crater size of 80µm. Ablated
particles were measured by a quadrupole ICP-MS (Micromass Platform) equipped
with a collision and reaction cell. Such a collision and reaction cell improves
carbonate analyses by eliminating interferences on mass 44. Scanned masses included
$^{24}$Mg, $^{26}$Mg, $^{27}$Al, $^{42}$Ca, $^{43}$Ca, $^{55}$Mn, $^{88}$Sr, $^{137}$Ba, $^{138}$Ba, $^{208}$Pb. Benthic foraminifera
from stations F (350 m) and E (552 m) were analyzed at ETH-Zurich (due to
laboratory renovations at Utrecht University). The laser type and ablation parameters
were identical to those detailed above. The ablated particles were measured using a
quadrupole ICP-MS (ELAN 6100 DRC, Perkin-Elmer). In both cases, calibration was
performed using an international standard (NIST610) with Ca as an internal standard



(Jochum et al. 2011). The same masses as measured in Utrecht were monitored, in
addition to $^7$Li, $^{23}$Na, $^{47}$Ti, $^{60}$Ni, $^{61}$Ni and $^{89}$Y. Inter-laboratory compatibility was
monitored using a matrix-matched calcite standard.

Analytical error (equivalent to 1 sigma), based on repeated measurement of an

external standard, was <5% for reported elements. Each laser ablation measurement
was screened for contamination by monitoring Al and Pb. On encountering surface
contamination, the data integration interval was adjusted to exclude any Al or Pb
enrichment. Cross-plots between Al and Pb versus Mn showed that they are unrelated,
confirming accuracy of the integrations.

During the laser ablation analyses the different trace elements were monitored

with respect to time, thus representing a cross section of the test wall. This allows not
only quantification of the different trace metals of interest, but also to observe
variability within individual tests. Each species has a distinct test-wall thickness,
permitting the study of intra-test variability. A typical ablation profile for *H. elegans*
is shown in Fig. 2.

**2.4 Analyses of manganese in foraminiferal tests**
Contamination and presence of secondary Mn-rich coatings on benthic foraminiferal
tests has been a longstanding challenge in trace metal analyses of benthic foraminifera
(Boyle 1983, Lea and Boyle 1989). In this study the trace metal data are based
exclusively on living (Rose Bengal stained) foraminifera, which effectively rules out
the impact of Mn-rich coatings on trace metal concentrations. At the time of sampling,
the collected tests were still enveloped by foraminiferal cytoplasm, preventing the
formation of extraneous inorganic precipitates. Although benthic foraminifera live
within the sediment, their test is physically separated from the environment as they



are enveloped in an organic sheath (Ní Fhlaithearta et al., 2013). In case a recently
deceased foraminifer was mistakenly analyzed (still with sufficient protoplasm to
stain with Rose Bengal) the Mn oxide would not only have had limited time to
develop, but it would also show up as a Mn spike at the start of a laser ablation profile.
The ablation profiles confirm that no Mn-rich phases are present at the test surfaces
(Fig. 2).

Comparing LA-ICP-MS data with traditional solution analyses for

foraminiferal Mg/Ca values showed that data are directly comparable (Rosenthal et al.,
2011). Also for trace metals such as $Ni^{2+}$, $Cu^{2+}$ and $Mn^{2+}$, cross-calibration of LA-
ICP-MS and micro-XRF shows those analytical results are robust (Munsel et al.,

2010).


## 217    2.5 Benthic foraminiferal Mn/Ca

Manganese incorporation in benthic foraminiferal test carbonate was analyzed from 4
different species (*Hoeglundina elegans, Melonis barleeanus, Uvigerina mediterranea,*
*Uvigerina peregrina*), from 6 coring sites, for up to 9 depths in the sediment. Sample
coverage for all stations is described in Table 2. Descriptive statistics are presented in
Table 3.

### 224    2.5.1 Intra-individual variability

From the largest taxon, *Uvigerina mediterranea*, 3-4 analyses were routinely carried
out per test, and no trend in Mn/Ca values was seen in consecutive growth stages.
From the other species two analyses were performed per test. The resolution of the
ablation profiles themselves does not allow quantifying changes in trace metals within
the test wall. Still, comparing the data within individual ablation profiles shows that

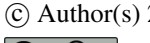



the intratest variability is generally limited for Mn (Table 4). As the ablation profiles
target one chamber mostly, this does not include the full potential range. Comparing
different ablation profiles between chambers in a single shell would circumvent this,
but this data is somewhat limited.

**2.5.2 Distribution characteristics of Mn/Ca in benthic foraminiferal calcite**
Boxplots are used to describe the range of Mn/Ca values and how the distribution,
median, average and skewness compares between species.. All ICP-MS
measurements are included, and as such represent both intra- and inter-individual
variation.

**3. Results**
**3.1 Pore water data**
Pore water dissolved manganese ($Mn^{2+}$) concentrations were measured at all six
stations. Manganese concentrations increase below the oxygen penetration depth at
stations C and D (Fig. 3), with the highest in-sediment $Mn^{2+}$ concentrations reached at
station D. At stations E and F manganese concentrations remain low after crossing the
oxygen penetration depth. At stations A and B the oxygen penetration depth and
$MnO_2/Mn^{2+}$ redox boundary are deeper than 10 cm's. Dissolved inorganic carbon
(DIC) and total alkalinity (TA), were measured at stations E, C and B (Fig. 4). At
stations D, C and E, DIC concentrations in the top 10 cm have a similar range (2350-
2700 µmol/kg). The DIC profile at station B has a narrower range, ranging from
2400-2550 µmol/kg. Total alkalinity values range from 3242 µmol/kg at station E to a
minimum of 2774 at station B. Carbonate ion concentrations $[CO_3^{2-}]$ were derived
based on TA and DIC values. The $[CO_3^{2-}]$ profiles were relatively similar (Fig. 4) for



stations E and C and B. Values for all three stations ranged from a maximum of 419
µmol/kg at station E to a minimum of 192 µmol/kg at station C (Fig. 4).

## 3.2 Mn/Ca data

### 3.2.1 Intra-individual variability

For most species some Mn/Ca analyses were below detection limit, except for *M.*
*barleeanus,* which contained measurable quantities of Mn in all shells analyzed. This
was most evident for *H. elegans*, where all but three Mn/Ca measurements were
below detection limit (dl). *Uvigerina peregrina* had a wider range of Mn/Ca values
than *U. mediterranea*. *Melonis barleeanus* exhibited the largest range of Mn/Ca
values of the four studied species (Fig. 5). For all species, except *H. elegans,* values
are somewhat skewed towards higher values.

### 3.2.3 Foraminiferal Mn/Ca variation across a depth transect

A trend of decreasing manganese incorporation with increasing water depth (350-
1987 m) is most clearly visible in *M. barleeanus* (Fig. 6), except that the maximum
values are observed at station E at 552 m. *Melonis barleeanus* shows the highest
Mn/Ca values and the largest Mn/Ca variability. Station E registers the broadest
Mn/Ca variability, which decreases with increasing water depth. *U. peregrina* also
exhibits the largest variability in Mn/Ca values at station E. For *U. peregrina*, $Mn^{2+}$
incorporation decreases from 350 m to 1987 m, except for station D (745 m), where
Mn/Ca values (between the 10 – 90th percentile) are approximately equivalent to those
at station A (350 m; Fig. 6). For *U. mediterranea* a trend of decreasing Mn
incorporation with increasing depth is found in specimens of *U. mediterranea* from
the sediments at 552, 745 and 980 m. The highest values are reached at the shallowest
station (350 m). Station E is also marked by the highest minimum Mn/Ca values for *U.*





*mediterranea*. At station A only two *U. mediterranea* measurements are above the
detection limit. *Hoeglundina elegans* shells from three stations (350 m, 1488 m and
1987 m) were analyzed, however, all but three measurements were below detection
limit (Fig. 6). These slightly elevated values were recorded at the shallowest station
(station F). These Mn/Ca values are still very low compared to ranges in Mn/Ca
values observed for the other species (Fig. 6).

Variability in Mn/Ca increases together with the overall Mn/Ca concentration

within benthic foraminiferal species (Table 4). This suggests that even at those
stations and depth levels where the highest Mn concentrations are recorded,
individuals with relatively low amounts of Mn in their calcitic test were found.
Comparing relative standard deviations, as a measure for the inter-specimen
variability, for the different stations and species suggests that with increasing Mn
concentration for *M. barleeanus* and *U. mediterranea* variability increases, whereas
for *U. peregrina* it decreases.

**3.2.4 In-sediment variation**
For most species Mn/Ca values are more or less constant with in-sediment depth (Fig.
3). However, *M. barleeanus* shows increasing Mn/Ca values with in-sediment depth.
This is most apparent at the shallowest station (station F - 350 m) (Fig. 3d ).

**4. Discussion**

Incorporation of Mn in benthic foraminiferal carbonate depends both on foraminiferal
ecology and early diagenesis in sediments. Although other factors such as temperature,
sea water carbonate chemistry, growth rate etc., might also affect the uptake of Mn in



the shell carbonate (Koho et al., 2017), these effects are most likely several orders of
magnitude smaller compared to the large range in dissolved Mn in pore water. Since
pore water Mn is the dominant factor controlling Mn incorporation, studies must
account for ecological controls, like foraminiferal depth habitat preference, as well as
for geochemical controls like oxygen concentrations and organic matter fluxes (Koho
et al., 2015; De Lange, 1986; Reichart et al., 2003).

## 4.1 Impact of redox conditions and foraminiferal habitat preference on Mn incorporation

In general, flux of organic matter arriving at the sea floor decreases with increasing
water depth, due to ongoing degradation during settling (Arndt et al., 2013 and
references theirein). Consequently, redox boundaries within the sediment generally
also deepen as a function of water depth, as oxygen consumption in the sediment
decreases. Such a fundamental organic matter-depth relation is in line with the much
deeper oxygen penetration depths at stations A and B compared to the more shallow
stations. At station F the relative shallow oxygen penetration depth observed is in line
with its' relative shallow water depth, although the organic matter which arrives here
at the seafloor apparently undergoes winnowing (Fontanier et al., 2008). The organic
matter along the transect studied is concentrated at a so-called depocenter, which
largely coincides with the depths of stations C and D (Fontanier et al., 2008). As
bottom waters at all stations are well oxygenated, organic matter concentration can be
considered the main control for redox conditions at stations F-A, with the amount of
organic matter arriving at the seabed being regulated by water depth and sedimentary
processes, such as focusing versus winnowing.



At stations C, D and F, the oxygen penetration depth and the $Mn^{2+}$ redox
boundaries are at the same depth, as expected. Station F shows the shallowest OPD of
all stations, although the organic matter concentration is relatively low. One
explanation for this observation is that a lower porosity at F (56% versus 76% and
79 % at stations D and E, respectively) impedes oxygen diffusion through the
sediment. Alternatively, the pore water profile reflects an earlier organic matter
deposition event, with this organic matter being largely consumed at the time of
sampling. The pore water profiles require more time to re-equilibrate to the new
conditions (Burdige and Gieskes, 1983). At station E there is a mismatch between
oxygen penetration depth and the $Mn^{2+}$ redox boundary as the $Mn^{2+}$ redox boundary is
considerably deeper than the OPD. Although this is in line with the observed higher
bioirrigation at this station (Fontanier et al., 2008), this might reflect non-equilibrium
conditions as well.
The vertical distribution of benthic foraminiferal species varies between
stations, in accordance with organic matter concentrations and redox zonation, which
is consistent with the TROX model (Jorissen et al., 1995; Fontanier et al., 2008). In
case of a shallower redox zone, infaunal benthic foraminifera biomineralize in contact
with Mn-enriched pore water, with highest dissolved manganese concentrations
occurring just below the oxygen penetration depth at all stations, except for station E
(552 m). This is in contrast to low bottom-water oxygen environments often studied in
the context of proxy development studies, where pore water $Mn^{2+}$ is released from the
pore water (Koho et al., 2015, 2017; Mangini et al., 2001).
The species studied here cover the range of shallow-infauna to intermediate-
infauna niches. *Hoeglundina elegans*, a typically shallow infaunal species, is often
found close to the sediment-water interface (Jorissen et al., 1998; Schönfeld 2001;





Fontanier et al., 2002; Fontanier et al., 2008) and contains the lowest concentration of
Mn in its test. Only at the shallowest station (350 m) three specimens of *H. elegans*
show concentrations above the detection level, with values still low compared to the
values observed for the other species (Fig. 6). In the Bay of Biscay Reichart et al.
(2003) also suggested that elevated Mn concentrations in *H. elegans* were confined to
stations with oxygen depleted bottom waters and/or with a shallow oxygen
penetration depth. *Uvigerina mediterranea* and *Uvigerina peregrina* are also classed
as shallow-infaunal species; they are typically found within the top few centimeters of
the sediment column (Fontanier et al. 2002, Fontanier et al. 2008). The calculated
average living depth ($ALD_{10}$) as calculated in Fontanier et al. (2008) is consistently
shallower than the $ALD_{10}$ for *U. mediterranea*. This is at odd with previous reports
suggesting *U. peregrina* has a slightly deeper microhabitat than *U. mediterranea*
(Fontanier et al. 2002; 2006). That *U. peregrina* has a deeper microhabitat is further
supported by the usual distinct $\partial^{13}C$ offset in *U. peregrine,* which is more depleted
compared to *U. mediterranea* (Schmiedl et al., 2004; Fontanier et al, 2002, 2006). The
higher Mn/Ca values observed here for *U. peregrina* (Figure 6) supports the idea that
it calcifies somewhat deeper in the sediment compared to *U. mediterranea*.
Alternatively, *U. peregrina* may migrate downwards within burrows to track food
resources, recording redox steepness (Loubere et al., 1995). This could highlight a
disparity between the assumed living depth (the depth interval of recovery) and
biomineralization depth of foraminifera. Still, this would also result in a higher
variability of Mn/Ca values at higher Mn/Ca levels, which is not observed. Hence,
more likely the observed disparity between the geochemical signals incorporated into
foraminiferal calcite and depth of recovery in *U. peregrina* reflects opportunistic
behaviour, with calcification at a shallower in-sediment depth in response to more



favourable conditions after e.g. seasonal peaks in organic matter fluxes (Accornero et
al., 2003), when the OPD is close to the sediment water interface.

*Melonis barleeanus*, generally considered an intermediate-infaunal species

(Fontanier et al., 2002, 2008), contains the highest concentrations of Mn in its test.
Manganese incorporation in this species increases with increasing labile organic
matter (Fig. 7a).

In summary, the habitat preference of the benthic foraminiferal species studied

here is reflected in the Mn/Ca values recorded in their tests. This is in contrast with
other results showing lower Mn/Ca values in foraminiferal tests with shallower redox
fronts (Koho et al., 2015). This, however, critically depends on the Mn being released
to the water column, which only occurs when the bottom waters are disoxic. In case
of a seasonal organic matter deposition event, an increase of Mn concentration in
foraminiferal test carbonate would initially occur in the deeper and ultimately also in
the more shallow calcifying foraminifera. This is in line with the conceptual
TROXCHEM[3] model, with the conditions studied here falling within the first stage of
the temporal succession considered in the model. Bottom water remains well
oxygenated ($O_2$ concentrations at the study area: 199-219 μmol/l (Fontanier et al.,
2008)) and organic matter loading is controlling $Mn^{2+}$ concentrations in the sediment.
To what extend species are high in Mn/Ca depends on living depth and opportunistic
behavior.

At a given location, a benthic foraminiferal species' depth preference or

biomineralization depth, is reflected in its average Mn/Ca value (Fig. 5). The trend
across a depth transect shows a strong correlation to labile organic matter
concentrations in the surface sediments (Fig. 7). The strong correlation between labile
organic matter (i.e. sedimentary lipid content) and Mn incorporation in shallow and





intermediate infaunal species *U. mediterranea* ($R^2 = 0.80$ ($p < 0.05$) suggests that test
Mn has potential as a proxy for detecting past labile organic matter fluxes. Notably,
*M. barleeanus* has a very strong correlation (0.81), though this correlation lacks
statistical significance ($p > 0.05$). In contrast, *U. peregrina* shows a correlation
coefficient of only 0.45 ($R^2$) between test Mn and labile organic matter. *Uvigerina*
*peregrina* is reported to respond opportunistically to the concentration and quality of
organic matter produced during bloom events (Fontanier et al., 2003; Koho et al.,
2008; Barras et al., 2010). This response is in the form of increased reproduction and
growth. Perhaps *U. peregrina* calcifies at shallow depths and therefore does not
capture the $Mn^{2+}$ gradient.
At low oxygen concentrations Mn is released through the reduction of
manganese (oxy)hydroxides. Here we show an increase in Mn/Ca incorporation in
several species, from shallow to intermediate-depth infaunal habitats, as a function of
oxygen penetration depth. Such a correlation agrees with studies by Ní Fhlaithearta et
al. (2010) and McKay et al. (2015) from a down core record of Mn/Ca$_{H. elegans}$ during
the formation of sapropel (S1) in the Eastern Mediterranean and a paleoproductivity
study of an upwelling system in the NE Atlantic, respectively. Here, a comparison of
Mn (oxy)hydroxides in the sediment and foraminiferal $Mn^{2+}$ showed that $Mn^{2+}$
incorporation in an epifaunal to shallow infaunal species was higher during times of
enhanced $Mn^{2+}$ remobilization and hence higher pore water $Mn^{2+}$. Such a correlation,
however, requires that the bottom waters remain somewhat oxygenated to retain the
dissolved $Mn^{2+}$ in the pore water. With disoxic bottom waters $Mn^{2+}$ escapes the
porewater and foraminiferal Mn/Ca values decrease (Koho et al., 2015). However,
with high organic matter deposition, which might be concentrated in events, also



foraminiferal species living at or close to the sediment water interface may show
elevated Mn concentrations.

In addition to the here observed changes, biomineralization could affect $Mn^{2+}$

incorporation. To date, however, no studies have been carried out under controlled
conditions to constrain species-specific offsets in Mn incorporation. In a controlled
laboratory study by Munsel et al (2010) Mn incorporation in *Ammonia tepida*
increased with increasing $Mn^{2+}$ concentrations in the culture water and the partition
coefficient was well above 1. The lack of any appreciable discrimination argues
against a major biomineralization impact on $Mn^{2+}$ partitioning. Also our data does not
suggest a major impact of biomineralization on Mn incorporation.

In summary, Mn incorporation seems primarily controlled by pore water

conditions in close proximity to the test, with a secondary control determined by the
ability of a foraminifer to seasonally calcify and migrate within the sediment.
**4.2 Pore water Mn dynamics and foraminiferal migration within the sediment**
Manganese is incorporated in foraminiferal carbonate with a partition coefficient (D)
close to 1 (Munsel et al., 2010), which argues for a minor biomineralization control.
We calculated Mn partition coefficients for *U. mediterranea*, *U. peregrina* and *M.*
*barleeanus* at stations E, C and B (Table 6) based on average $Mn/Ca_{foram}$ and average
$Mn/Ca_{pore\ water}$ values found above the $Mn^{2+}$-MnO(H) redox boundary. Calculated $D_{Mn}$
agrees with the previously reported $D_{Mn}$ by Munsel et al., (2010), with values varying
between ~1-2 for *U. mediterranea* and *U. peregrina*. The Mn partition coefficient for
*Melonis barleeanus* ranges from ~4-7. The partition coefficient for this species most
likely reflects its capacity to calcify under dysoxic conditions, close to or even below
the oxygen penetration depth. Still, this calculation is based on two assumptions: (1)
the depth foraminifera are recovered from during sampling corresponds with the



average depth of calcification and, (2) variation in pore water is limited. Establishing
species specific Mn partitioning coefficients using culture experiments might,
however, be needed for unlocking the full potential of this proxy.

A foraminifer calcifying within a steep $Mn^{2+}$ gradient is exposed to a higher

range of $Mn^{2+}$ concentrations (over a fixed depth interval) compared to specimens
living along a more gradual $Mn^{2+}$ concentration gradient. Since foraminifera can
migrate through the sediment as a response to food availability and oxygen
concentrations (Alve and Bernhard 1995; Gross, 2000), not only the slope of the Mn
gradient, but also the in-sediment depth range (microhabitat) of the foraminifer in
relation to the Mn redox boundary, should be considered (Fig. 8). A shallow-infauna
species, with a limited in-sediment range, would be expected to exhibit lower
variability than an intermediate- infauna species, which possibly migrates
considerably in depth. This is exemplified at station F (350 m) where we note an
increase in foraminiferal test Mn/Ca variability at 2 cm depth, consistent with the
oxygen penetration depth at that station (Fig. 3). Moreover, the variability in Mn/Ca
values increases towards higher Mn/Ca values. This is in line with *M. barleeanus*
traveling more actively through the redox zones than *U. mediterranea* or *U. peregrina*.
Nitrate respiration could be mechanism allowing this dynamic behaviour by *M.*
*barleeanus* in the intermediate depth habitat. However, Pina-Ochao et al. (2010),
studying denitrification in foraminifera, reports nitrate storage in all three species
mentioned here. Notably, nitrate storage in *M. barleeanus* is lower than *U.*
*mediterranea* and *U. peregrina*. Alternatively *M. barleeanus* thrives in habitats with
varying oxygenation and hence also varying Mn levels, whereas the stable but high
Mn/Ca values in the Uvigerinids are related to their opportunistic behaviour.





With a redox-sensitive element such as Mn, in a dynamic geochemical
environment, it is not surprising that foraminifera exhibit high inter-individual
variability in their Mn/Ca incorporation. Benthic foraminifera reside in a 3D
geochemical mosaic, as reflected by a large spread of Mn values, in addition to
undergoing substantial temporal variability. Still, using Mn/Ca as a potential proxy
for redox conditions or primary productivity seems promising, as established
ecological characteristics of species are reflected by differences in Mn incorporation.
Apparently the large variability on both spatial and temporal scales averages out,
making Mn into a promising proxy for paleo-redox and organic matter flux.

**5. Conclusion**
This study investigates the link between benthic foraminiferal habitat preferences and
manganese incorporation in their tests. Manganese incorporation increases with
bottom-arriving labile organic matter content, driven by enhanced oxygen demand.
This results in a more shallow oxygen penetration depth with immediately below it
enhanced dissolved Mn levels. Shallow infaunal species calcify under lower
concentrations of Mn compared to intermediate infauna, in line with their depth
preference. Their depth habitat is related to in-sediment changes in redox conditions.
However, these distribution not necessarily vary synchronous with changes in redox
zonation as illustrated by the Mn/Ca variability in their tests (Fig. 8). The latter
reflects the Mn/Ca porewater composition, which itself is directly related to reactive
organic matter concentration and redox conditions. The foraminiferal Mn/Ca ratio and
inter-specimen variability, therefore, provides information on past Mn cycling within
the sediment. Consequently, the foraminiferal Mn/Ca ratio is a potential proxy for
bottom-water oxygenation and organic matter fluxes.




**Acknowledgements**
We thank captain and crew of the N/O Téthys 2 (CNRS-INSU) for their assistance
during the *BEHEMOTH* campaign. We acknowledge the technical assistance given by
Christine Barras, Mélissa Gaultier, Sophie Terrien and Gérard Chabaud from Angers
and Bordeaux University. We thank Serge Berné and Laetitia Maltese (Ifremer), for
providing us with maps of the study area and Xavier Durrieu de Madron (Perpignan
University) for discussions about water column structure. Helen de Waard (LA-ICP-
MS) and Karoliina Koho (SEM) (Utrecht University) are acknowledged for their
laboratory assistance. The Darwin Center for Biogeosciences provided partial funding
for this project. This paper contributes to the Netherlands Earth Systems Science
Center (NESSC –www.nessc.nl)

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





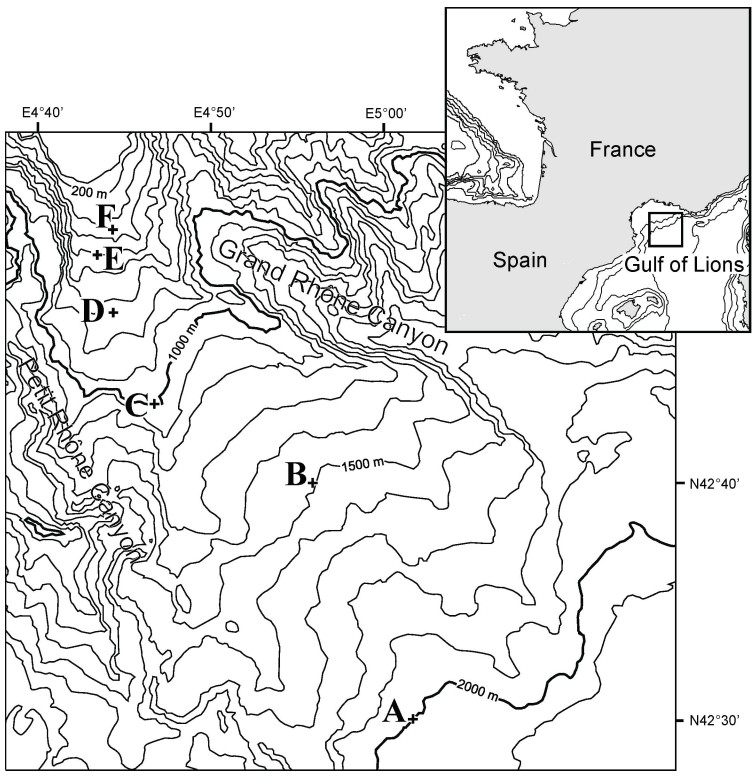

**Figure 1. Location map showing sampling stations and bathymetry.**





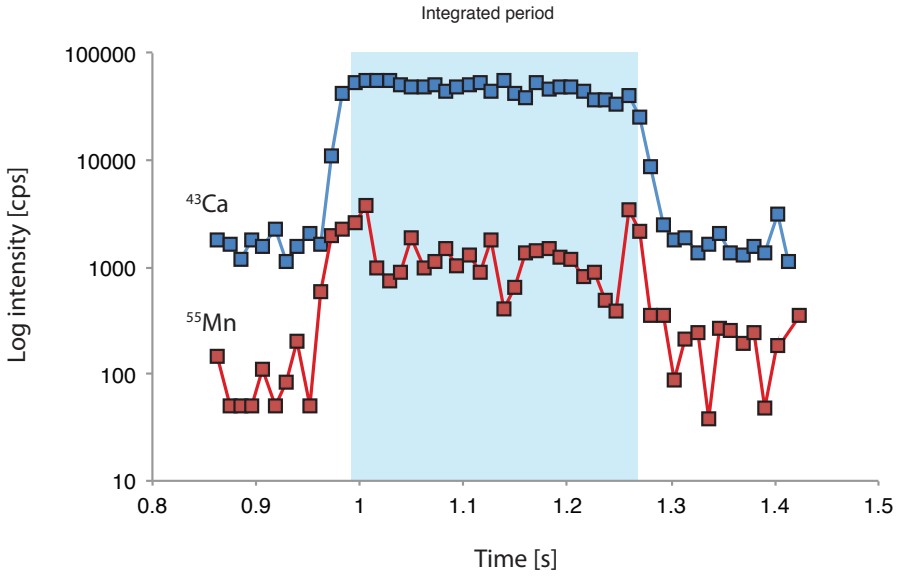

Figure 2. Example of a laser ablation profile, signal log intensity counts per second [cps] through time. The integrated signal is shaded.



Figure 3. Plots of Mn/Ca (µmol/mol) measured in living (stained) Hoeglundina elegans, Uvigerina mediterranea, Uvigerina peregrina and Melonis barleeanus. Individual analyses are plotted (grey circles) alongside average values for a given depth in the sediment (red squares). Porewater Mn2+ (µmol/kg) profiles (black line) are plotted for all stations. The dashed grey line indicates the oxygen penetration depth (OPD).





Figure 4.

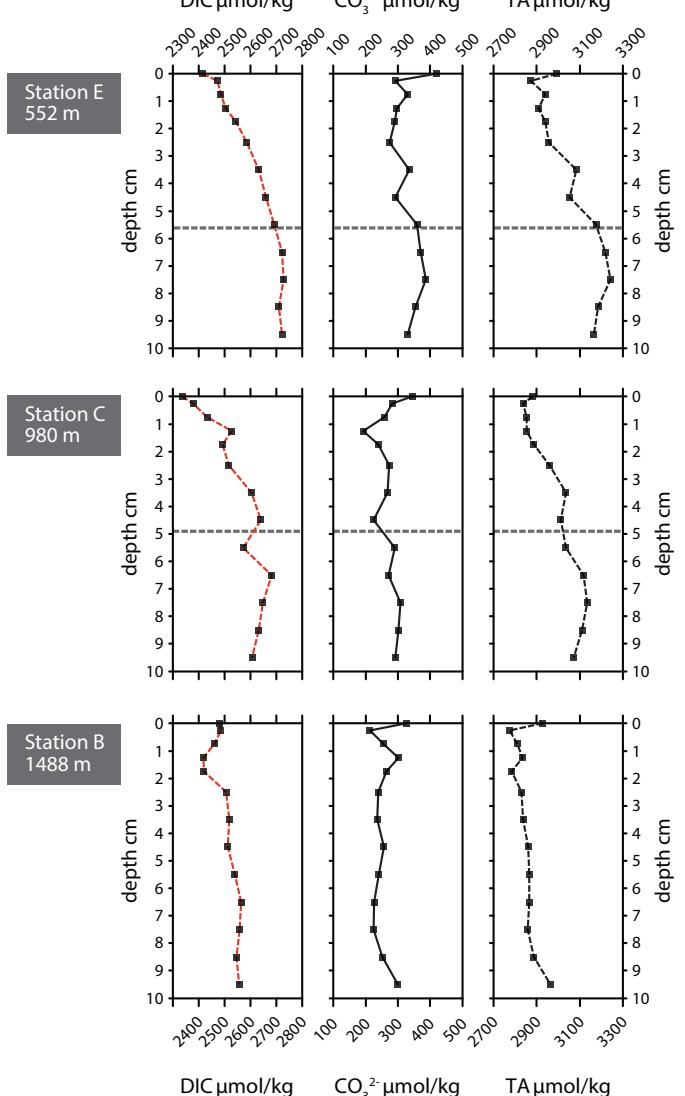

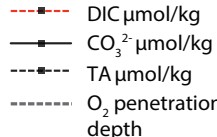

Figure 4. Carbonate chemistry parameters for stations E, C and B. A) Dissolved inorganic carbon (DIC), B) [CO32-] in µmol/kg, C) Total alkalinity (TA) in µmol/kg.



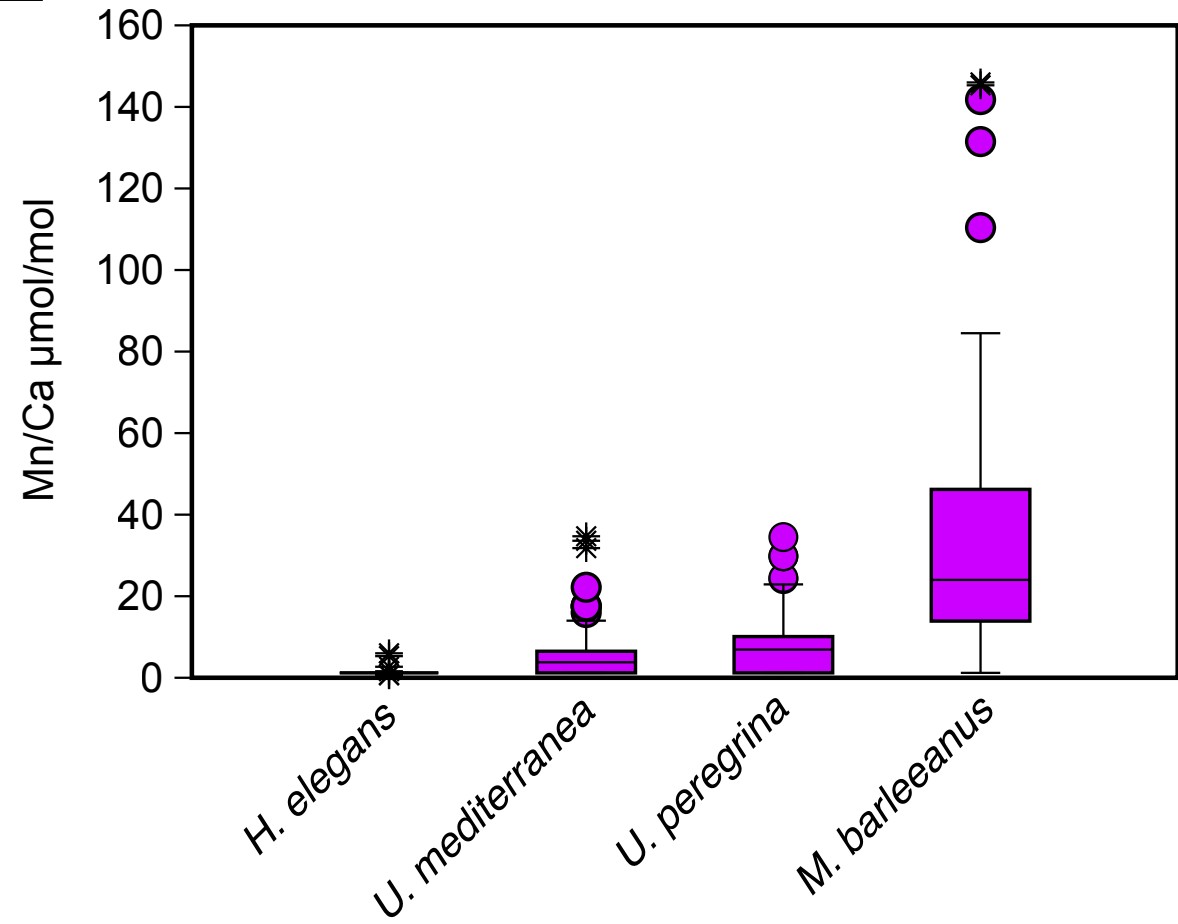

Figure 5. Box plots describing the distribution in Mn/Ca values measured in living (stained) individuals of Hoeglundina elegans, Uvigerina mediterranea, Uvigerina peregrina and Melonis barleeanus. The box represents all values between the 25th and 75th percentile. The dissection line through the box denotes the median. The whiskers are drawn from the top of the box up to the largest data point less than 1.5 times the box height from the box and similarly below the box. Values outside the whiskers are shown as circles, values further than 3 times the box height are denoted as stars.



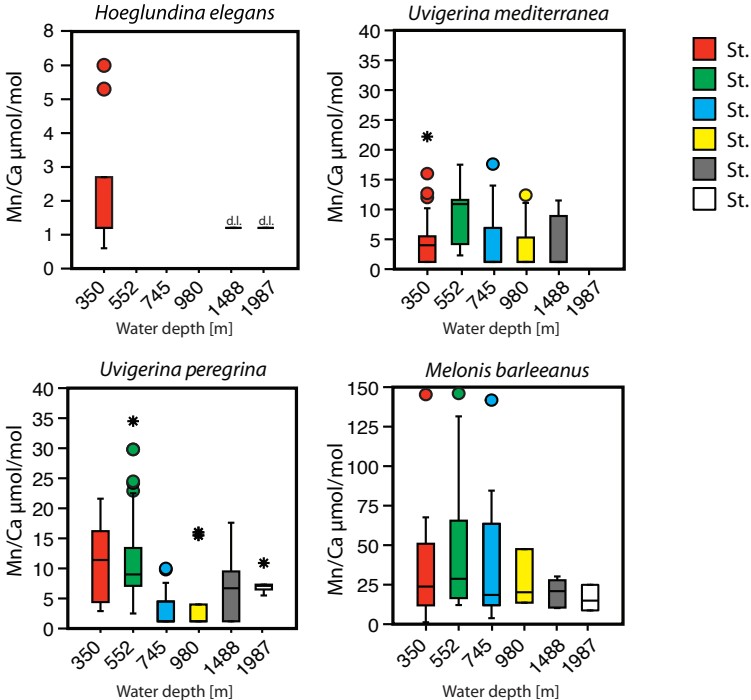

Figure 6. Box plots describing the distribution of Mn/Ca values across a depth transect (350-1987 m) measured in living (stained) individuals of Hoeglundina elegans, Uvigerina mediterranea, Uvigerina peregrina and Melonis barleeanus. Note that the scale of the y-axis varies. The box represents all values between the 25th and 75th percentile with the whiskers extending less than 1.5 times the box height. The dissection line through the box denotes the median. Values outside the whiskers are shown as circles, values further than 3 times the box height are denoted as stars.





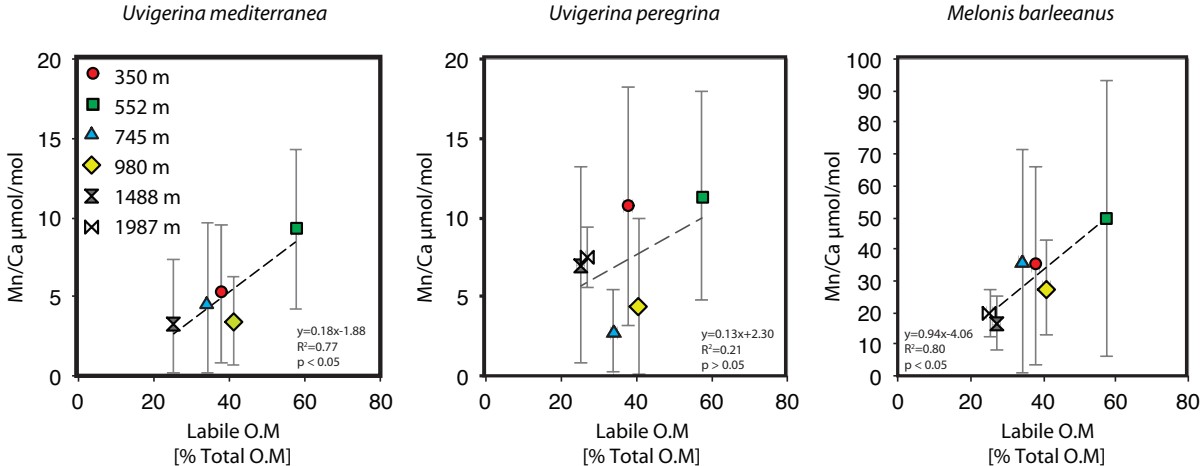

Figure 7. Plots of average Mn/Ca µmol/mol versus labile organic matter [% Total Organic matter].




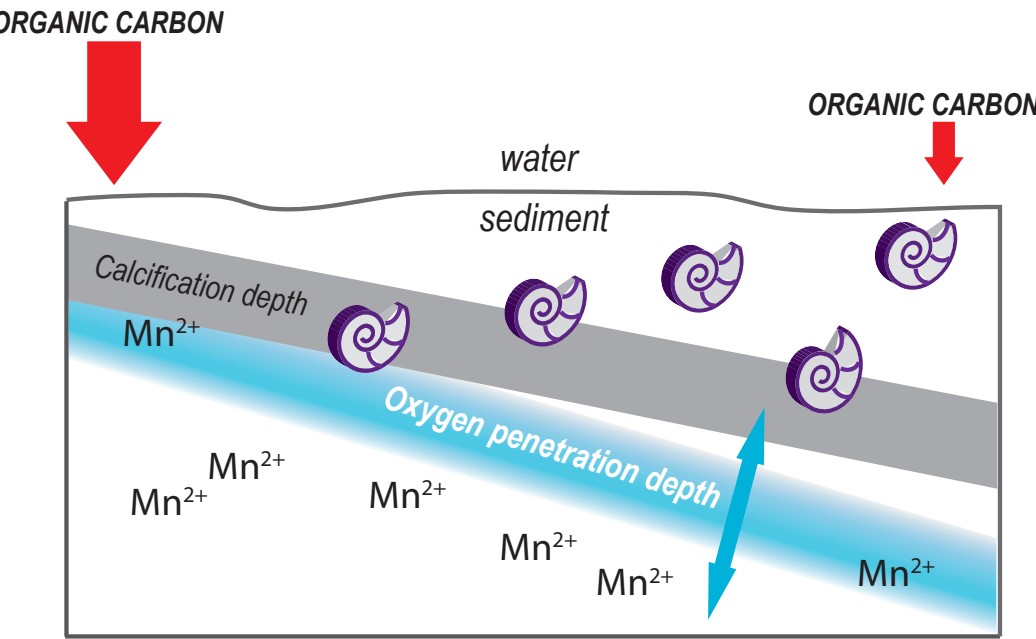

Figure 8. Schematic diagram of a shallow-infaunal and intermediate-infaunal benthic foraminifera and their spatial relationship with the sediment redox boundaries, migration zone, ALD and calcification depth.





Table 1. Water depth, coordinates and bottom water physio-chemical parameters: Temperature (°C), salinity, and oxygen penetration depth mm) for six stations F-A. (* Xavier Durrieu de Madron, Pers. Comm.)

| Station | Depth (m) | Latitude (N) | Longitude (E) | Bottom water temperature* (°C) | Bottom water salinity * | Oxygen penetration depth (mm) |
|---------|-----------|--------------|---------------|-------------------------------|------------------------|-------------------------------|
| F | 350 | 42°52'32 | 4°42'43 | 13.2 | ~38.5 | 20.5 ±3.3 |
| E | 552 | 42°48'78 | 4°43'21 | 13.2 | ~38.5 | 57.2 ±4.5 |
| D | 745 | 42°46'66 | 4°43'91 | 13.1 | ~38.5 | 36.5 ±1.6 |
| C | 980 | 42°43'18 | 4°46'58 | 13.1 | ~38.48 | 50.7 ±6.3 |
| B | 1488 | 42°38'83 | 4°56'03 | 13.1 | ~38.46 | 141.5 ±0.0 |
| A | 1987 | 42°28'25 | 5°00'61 | 13.1 | ~38.46 | 197.0 ±11.0 |



Table 2. Number of LA-ICP-MS analyses per benthic foraminifera species per sample per station.

| Station | Depth (m) | Sample intervals (cm) | *Hoeglundina elegans* no. analyses | *Uvigerina mediterranea* no. analyses | *Uvigerina peregrina* no. analyses | *Melonis barleeanus* no. analyses |
|---|---|---|---|---|---|---|
| **F** | 350 | 0-0.5 | 2 | 18 | 5 | 1 |
| | | 0.5-1 | 4 | 13 | | 2 |
| | | 1-1.5 | | 3 | | |
| | | 1.5-2 | | | | 10 |
| | | 2-2.5 | 1 | | | 2 |
| | | 3-3.5 | | | | 3 |
| | | 5-6 | | | | 3 |
| **E** | 552 | 0-0.5 | | 5 | 26 | |
| | | 0.5-1 | | | 9 | 4 |
| | | 1-1.5 | | | 14 | 3 |
| | | 1.5-2 | | | 5 | 3 |
| | | 2-2.5 | | | 7 | 3 |
| | | 2.5-3 | | | 6 | 2 |
| | | 3-3.5 | | | 6 | |
| | | 3.5-4 | | | 8 | 1 |
| | | 4-5 | | 3 | | |
| | | 7-8 | | | | 1 |
| **D** | 745 | 0-0.5 | | 20 | 13 | |
| | | 0.5-1 | | 6 | | 5 |
| | | 1-1.5 | | 3 | 6 | |
| | | 1.5-2 | | 7 | 8 | 6 |
| | | 2-2.5 | | | 4 | |
| | | 3.5-4 | | | 2 | |
| | | 4-5 | | | 2 | 4 |
| | | 8-9 | | | 2 | |
| **C** | 980 | 0-0.5 | | 20 | | 2 |
| | | 0.5-1 | | 20 | | 2 |
| | | 1-1.5 | | 4 | | |
| **B** | 1488 | 0-0.5 | 3 | 4 | 10 | 4 |
| | | 0.5-1 | 9 | 5 | 3 | 5 |
| **A** | 1987 | 0-0.5 | 15 | | 10 | |
| | | 1-1.5 | | | | 3 |



Table 3. Descriptive statistics (minimum, maximum, mean, median, standard deviation and interval of maximum frequency of total analyses for *H. elegans*, *U. mediterranea*, *U. peregrina* and *M. barleeanus* for Mn/Ca µmol/mol.

| Mn/Ca µmol/mol | *H. elegans* | *U. mediterranea* | *U. peregrina* | *M. barleeanus* |
|---|---|---|---|---|
| Min | dl* | dl* | dl* | 3.91 |
| Max | 0.69 | 22.71 | 35.38 | 149.50 |
| Mean | 0.04 | 4.03 | 8.28 | 37.22 |
| Median | dl* | 1.04 | 7.45 | 24.76 |
| Std. deviation | 0.16 | 5.03 | 7.17 | 35.17 |
| Max. frequency interval | dl-7.50 ( 100% < 1) | dl-7.50 (80%) | dl-7.50 (53%) | 7.5-15 (23%) |





Table 4. Relative standard deviation (% RSD) of intra-individual values in Mn/Ca within four species of benthic foraminifera (*H. elegans*, *U. mediterranea*, *U. peregrina* and *M. barleeanus*).

| Element | *H. elegans* % RSD | *U. mediterranea* % RSD | *U. peregrina* % RSD | *M. barleeanus* % RSD |
|---|---|---|---|---|
| Mn | 21 | 23 | 20 | 51 |



Table 5. Relative standard deviation (% RSD) of inter-individual values in Mn/Ca within four species of benthic foraminifera (*H. elegans*, *U. mediterranea*, *U. peregrina* and *M. barleeanus*).

| Element | *H. elegans* % RSD | *U. mediterranea* % RSD | *U. peregrina* % RSD | *M. barleeanus* % RSD |
|---|---|---|---|---|
| Mn | 400 | 125 | 87 | 97 |





Table 6. Manganese porewater – carbonate partition coefficient for foraminiferal species *Uvigerina mediterranea*, *Uvigerina peregrina* and *Melonis barleeanus*.

| Station | Partition coefficient (D)[1] | | |
| --- | --- | --- | --- |
| | *U. mediterranea* | *U. peregrina* | *M. barleeanus* |
| E (552 m) | 1.7 | 1.8 | 7.0 |
| C (980 m) | 1.2 | - | 5.1 |
| B (1488 m) | 2.2 | 2.3 | 4.1 |

[1]Porewater-carbonate partition coefficients were calculated using the average porewater Mn/Ca [µmol/mol] measured above the oxygen penetration depth and the average carbonate Mn/Ca [umol/mol] measured in *U. mediterranea*, *U. peregrina* and *M. barleeanus*, for all specimens recovered above the reported oxygen penetration depth (Fontanier et al., 2008).