# Peer review of "Manganese incorporation in living (stained) benthic foraminiferal"

_Biogeosciences, 2018_

## Referee Comment (RC1) · Anonymous Referee #1 · 2 Apr 2018

This study looks at the potential for the Mn/Ca of benthic foraminiferal calcite to act as a proxy for pore water oxygenation and labile organic matter. In order to do this, the authors have measured the pore water chemistry as well as the Mn/Ca geochemistry of 4 species of benthic foraminifera from a depth transect of cores in the Mediterranean. Analysis focuses on the living foraminifera recovered from the upper 10 cm of each of these core sites. Ni Fhlaithearta et al., find that the Mn/Ca of benthic foraminifera reflects the pore water environment from which they calcified and the flux of organic material to the site. However, the amount of incorporation and variability in Mn/Ca

incorporation is governed by species specific KD as well as ecological and depth preferences in addition to environmentally controlled pore-water conditions.

Overarching comments/questions: The authors analyzed select foraminifera species from specific core depths, however much of the article relies on speculation and inferences from existing literature as to the calcification or habitat depth of these same species. This is clearly extremely relevant to the interpretation of any pore water proxy. Is there a reason that species abundances with depth (at least for the relevant species) has not been included here? It would seem that inclusion of this could clarify some questions of habitat preference, and the degree to which this varies between sites.

It seems to me that the greatest barrier to application of these results to the fossil record is the issue of preservation. The authors discuss this clearly. However, I wonder if the research could not be even more impactful with a statement as to how this could be circumvented (at least in some environments). For example, did the authors undertake any comparative analyses of non-living and potentially altered specimens from the same cores? Could Mn-rich coatings be identified using the same LA techniques applied here to living foraminifera?

27 – and 52-53 – "surface and bottom water" – consider rephrasing surface (pelagic or near-surface?), as most planktic foraminifera do not actually reflect surface conditions 262 – What was the detection limit and how was it established? 367 – peregrina 407 – what is the p value? If the correlation is not significant, is it still meaningful? Also for Fig. 7, can you include the p values? 412-413- where was U. peregrina actually found in these sample? In relation to U. mediterranea?

---

## Referee Comment (RC2) · Anonymous Referee #2 · 27 May 2018

This manuscript aims to understand Mn incorporation into benthic foraminifera and explore its potential use for reconstructing pore water redox condition and organic matter content of sediments. Although the topic is potentially very interesting I have serious concerns about the analytical side of Laser ablation measurements. The authors should clarify these issues before 'interpretation/discussion' part of the manuscript can be evaluated. Therefore, I recommended major revision for this work. Below, I summarised the questions for the analytical part of the work.

1- Detection limits for Mn/Ca measurements. Mn values in ontogenetic (i.e. not altered)

[Figure]

foraminiferal calcite is very low (umol/mols) which make it challenging to accurately quantify with laser ablation measurements. Usually in our lab we use large laser spot size and energy to get sufficient signal to noise ratio (>100). The authors in this study provided very basic description of analytical procedures in the method section, which overshadows the result and discussion as there is no assurance on the quality of the measurements. The main concern I have is lack of any estimation on detection limits of their method. Fig 2 shows typical ablation profile BUT the Mn signal to noise ratio is very low (<10). Such low noise to signal ratios usually correspond to very noisy measurements (large error bars), which in fact is a common feature of the data presented in this work (figs 5,6,7). In fig 6a, there are labels 'LD' which I presume indicate detection limits and they are 1umol/mol. If this is true detection limit then majority of the data presented in this work (in exception of data for Melonis) has very little analytical base. Simply it is too close to detection limits compared to error bars and therefore statistically indistinguishable from noise. The authors should really accurately estimate their errors in the background (LD = 2SD of the variance in the background signal) and also variance in the signal itself. This is crucial for interpretation of the data. For example, the summary in fig 7 as it presented now shows no trends as the errors are huge and horizontal line is the best solution for these plots. Note, if 1 umol/mol is the detection limit, then more than 60% of the data is within the error bar from the noise.

2- Inappropriate standards for calibration. The authors used NIST610 for Mn/Ca calibration. This standard has ∼400ppm of Mn, which is >10,000 times higher than typical foraminifera values. It is advisable to use Nist 612,614 pair for this kind of application to avoid artefacts/noise in calibration. The typical LA-ICPMS will give 3-5% reproducibility on NIST glass. Considering that calibration is one point calibration and 0=blank, the 5% variability at 400ppm will result in large variance at few ppbs level. Considering very low noise/signal ratio (see above) all propagated errors will case huge variance in the resulting data. I am afraid this has to be fixed before discussing the science behind Mn incorporation into foraminifera.

Minor comments

- Measurements in the lab should be validated by their lab publication/s therefore sections 211-215 cannot assure the accuracy of the measurements. It has been mentioned in lines 181-182 that calcite standard were analysed for consistence. Data for this reproducibility will be the best indication of accuracy and reproducibility of the method and has to be reported.

- Section 2.5/2.5.1/2.5.2 (lines 217-239). It is necessary to break this down in sections if there are only 2 sentences in each section?

---

## Author Response (AR1)

Dear Editor,

We would like to thank you for your comment. In your comment you point out a common misconception, in which the depth at which organisms are recovered is mistakenly interpreted as also being the depth at which they calcify. As you rightfully indicate this is not necessarily so. This is also one of the main rationales underlying our study and we hoped that this is clear throughout the manuscript. Already at lines 67 and 68 of the original manuscript we hence tried to highlight the difference between living depth and calcification depth. This sentence is somewhat rephrased now to make this more clear and now reads: "Linking foraminiferal test chemistry with pore water chemistry requires in-depth knowledge of, 1) how early diagenesis in sediments affects pore water chemistry, 2) the habitat preference of the foraminiferal species, 3) foraminiferal migration (and the depth at which they calcify) within the uppermost sediment layer."

You also pointed out the differences in living depth between *Uvigerina peregrina* and *U. mediterranea*. At lines 366 and 368 we referred to *U. mediterranea* living deeper than *U. peregrina*, similar to your comment. This is now emphasized in the revised version at lines 359-361: "Both *Uvigerina mediterranea* and *M. barleeanus* were in the Gulf of Lyons found to occupy shallow to intermediate infaunal habitat, with *U. peregrina* having a somewhat shallower infaunal habitat (Fontanier et al., 2008)". This is based on existing literature and counts on living foraminifera, which we do not dispute. We fully agree that the difference observed is more likely due to a difference in calcification depth. We indicated this at lines 372-374 of the original manuscript. See also lines 412-413 of the original manuscript, pertaining to the same.

For *Melonis barleanus* Mn uptake is fully in line with its deeper habitat, and depth habitat and calcification depth probably coincide. See also lines 386-387 and 450-452 of the original manuscript. Although we suggest that *M. barleanus* potentially "travels" more along a depth gradient in the sediment (lines 469-470), this would still be consistently deeper than the depth habitats of both *U. peregrina* and *U. mediterranea*. In the revised manuscript this is therefore now changed to: "This is in line with the depth habitat of *M. barleeanus* being consistently deeper and this species traveling more actively through the redox zones than *U. mediterranea* or *U. peregrina*."

In the revised version we now also added at lines 390-394: "*Melonis barleeanus*, generally considered an intermediate-infaunal species (Fontanier et al., 2002, 2008), contains the highest concentrations of Mn in its test, which is in line with the deepest habitat of the species studied here", to better reflect this fact.

We hope that our answers and the additional changes convince the editor to forward our manuscript.

Many thanks, also on behalf of all other co-authors,

Gert-Jan Reichart

Original review:

Dear Editor,
We are glad to have received two helpful sets of comments. Below we have copied the reviewer's comments one at the time and indicate how we have addressed them or (in a few cases) argue why we respectfully disagree. We would like you to consider the revised manuscript for publication in Biogeosciences.

**Anonymous Referee #1**

This study looks at the potential for the Mn/Ca of benthic foraminiferal calcite to act as a proxy for pore water oxygenation and labile organic matter. In order to do this, the authors have measured the pore water chemistry as well as the Mn/Ca geochemistry of 4 species of benthic foraminifera from a depth transect of cores in the Mediterranean. Analysis focuses on the living foraminifera recovered from the upper 10 cm of each of these core sites. Ni Fhlaithearta et al., find that the Mn/Ca of benthic foraminifera reflects the pore water environment from which they calcified and the flux of organic material to the site. However, the amount of incorporation and variability in Mn/Ca incorporation is governed by species specific KD as well as ecological and depth preferences in addition to environmentally controlled pore-water conditions.

Overarching comments/questions: The authors analyzed select foraminifera species from specific core depths, however much of the article relies on speculation and inferences from existing literature as to the calcification or habitat depth of these same species. This is clearly extremely relevant to the interpretation of any pore water proxy. Is there a reason that species abundances with depth (at least for the relevant species) has not been included here? It would seem that inclusion of this could clarify some questions of habitat preference, and the degree to which this varies between sites.

Author's response: As referee 1 states, the species abundances with depth have not been added to this manuscript as they were published before already. Included in the manuscript is a brief reference to the average living depth (ALD$_{10}$), for *U. mediterranea* and *U. peregrina* (line 363-364). The species abundances in these samples have previously been reported in Fontanier et al., (2008). We are aware that it was recently published that partitioning with respect to Mn may vary between species (Barras et al., in press). This was not known at the time the discussion paper was submitted. We now added this to the discussion and refer to the recent paper by Barras et al. (in press) and added to the text:
New Lines 429-434: "Recently Barras et al. (2018), also using controlled growth experiments, showed however that Mn partitioning in B. marginata differs from that in *A. tepida*, with that in *B. marginata* being close to one and that of A. tepida being 4 times lower. Inter-specific differences are considerable and hence an impact of biomineralization on Mn incorporation can not be disregarded."

Authors's changes to the manuscript: A summary of the species abundances, based on Fontanier et al., (2008) table 5 of that paper, has been added to the text at lines 348-350: 'Both *Uvigerina mediterranea* and M. barleeanus were in the Gulf of Lyons found to occupy shallow to intermediate infaunal habitat, with *U. peregrina* having a somewhat shallower infaunal habitat (Fonatnier et al., 2008).'

It seems to me that the greatest barrier to application of these results to the fossil record is the issue of preservation. The authors discuss this clearly. However, I wonder if the research could not be even more impactful with a statement as to how this could be circumvented (at least in some environments). For example, did the authors undertake any comparative analyses of non-living and potentially altered specimens from the same cores? Could Mn-rich coatings be identified using the same LA techniques applied here to living foraminifera?

Author's response: One way to circumnavigate issues with preservation of the original Mn signal and hence the possibility to apply our approach to the fossil record, is by specifically targeting *H. elegans* for Mn/Ca measurements. The aragonitic nature of this organism prevents overgrowths. Such an approach was previously applied for the reconstruction of the paleo-environment during deposition sapropel S1 in the Mediterranean (Ni Fhlaithearta et al., 2010).
For the other species (*Uvigerina mediterranea, Uvigerina peregrina* and *Melonis barleeanus* ) studied in this paper it would potentially be possible to analyse fossil specimens using selection criteria (for example, degree of visible alteration, diagenetic Fe incorporation, etc.). With laser ablation measurements it is furthermore possible to adjust the analytical window in such a way as to exclude diagenetic coatings (based on the trace metal signature within the test wall), which has been described in several papers (e.g. Reichart et al., 2003).

Author's changes in the manuscript: We added to the discussion a sentence describing this potential approach at lines 307-312 'The fact that this study was based on living foraminifera circumvents potential complications due to Mn-rich coatings. Such coatings would likely not affect the aragonitic shell of *H. elegans* (Ní Fhlaithearta et al., 2010), but might interfere when analyzing fossil calcite shells. Still, a spatially resolved analytical technnique like LA-ICP-MS allows detecting such coatings also in fossil specimens.'

– and 52-53 – "surface and bottom water" – consider rephrasing surface (pelagic or near-surface?), as most planktic foraminifera do not actually reflect surface conditions

Author Response: No reference to 'surface and bottom water' is seen at line 27. In line 52-53, 'surface' has been changed to 'pelagic'.
Author's changes to the manuscript: 52-53: 'Both pelagic and bottom water conditions...'

– What was the detection limit and how was it established?

The detection limit differs for each ablation and is a combination of the number of scans collected (acquisition time) and the background for a certain element. These levels are hence calculated using data reduction software for each single ablation profile. For Mn/Ca in the foraminiferal shells analysed here detection limits were on the order of 1.2 umol/mol. This is indicated in figure 6 of the manuscript.

– peregrina
Author Response: changed accordingly.

– what is the p value? If the correlation is not significant, is it still meaningful? Also for
Fig. 7, can you include the p values?

The exact p-values is not calculated. The fact that it is over 0.05 indicates that, in view of the number of samples, it is not significant. This is stated in the manuscript. The correlation is indeed included as the high value hint towards a relationship. This is included because although a high p-value implies that a statistical significant correlation cannot be proven, it does not exclude a relation may still exist.

412-413- where was U. peregrina actually found in these sample? In relation to U. mediterranea?

Uvigerina mediterranea has indeed been classified as intermediate to shallow infaunal and *U. peregrina* as shallow infaunal at this location (Fontanier et al., 2008). The somewhat different incorporation of Mn in U. peregrina indeed suggests that it calcifies somewhat more shallow, which is in line with the microhabitat study. This is now clarified in the revised manuscript (lines 416-417).

**Anonymous Referee #2**

This manuscript aims to understand Mn incorporation into benthic foraminifera and explore its potential use for reconstructing pore water redox condition and organic matter content of sediments. Although the topic is potentially very interesting I have serious concerns about the analytical side of Laser ablation measurements. The authors should clarify these issues before 'interpretation/discussion' part of the manuscript can be evaluated. Therefore, I recommended major revision for this work. Below, I summarised the questions for the analytical part of the work. 1- Detection limits for Mn/Ca measurements. Mn values in ontogenetic (i.e. not altered)foraminiferal calcite is very low (umol/mols) which make it challenging to accurately quantify with laser ablation measurements. Usually in our lab we use large laser spot size and energy to get sufficient signal to noise ratio (>100). The authors in this study
provided very basic description of analytical procedures in the method section, which overshadows the result and discussion as there is no assurance on the quality of the measurements.

We agree with the reviewer that the analyses of Mn in foraminiferal test is analytically challenging. We have invested much effort in improving this type of analyses since they were published for the first time (Reichart et al., 2003). Currently we are able to not only analyse Mn/Ca in benthic foraminifera, but are now also able to reliably analyse Mn/Ca in planktonic foraminifera (e.g. Steinhardt et al., 2014). The analytical procedures have been described in detail in several publications, which we refer to (e.g. Koho et al., 2015; 2017). Matrix matched standards were used to verify the analytical procedures and consistency.

The main concern I have is lack of any estimation on detection limits of their method. Fig 2 shows typical ablation profile BUT the Mn signal to noise ratio is very low (<10). Such low noise to signal ratios usually correspond to very noisy measurements (large error bars), which in fact is a common feature of the data presented in this work (figs 5,6,7). In fig 6a, there are labels 'LD' which I presume indicate detection limits and they are 1umol/mol. If this is true detection limit then majority of the data presented in this work (in exception of data for Melonis) has very little analytical base. Simply it is too close to detection limits compared to error bars and therefore statistically indistinguishable from noise. The authors should really accurately estimate their errors in the background (LD = 2SD of the variance in the background signal) and also variance in the signal itself. This is crucial for interpretation of the data. For example, the summary in fig 7 as it presented now shows no trends as the errors are huge and horizontal line is the best solution for these plots. Note, if 1 umol/mol is the detection limit, then more than 60% of the data is within the error bar from the noise.

We are very much aware of the fact the analyses are close to the detection limit. For each profile the detection limit was calculated already according to the procedure suggested by the reviewer. We also know that part of the inter species variability will be due to analytical issues, which is exactly why we have discussed this as a separate issue. We have tried to rephrase this somewhat to accommodate the referees concern. Lines 480-482: 'Although the analyses of foraminiferal test Mn/Ca is challenging, which adds to the inter-specimen variability, we observe systematic differnces between species in Mn/Ca variability.'.

2- Inappropriate standards for calibration. The authors used NIST610 for Mn/Ca calibration. This standard has _400ppm of Mn, which is >10,000 times higher than typical foraminifera values. It is advisable to use Nist 612,614 pair for this kind of application to avoid artefacts/noise in calibration. The typical LA-ICPMS will give 3-5% reproducibility on NIST glass. Considering that calibration is one point calibration and 0=blank, the 5% variability at 400ppm will result in large variance at few ppbs level. Considering very low noise/signal ratio (see above) all propagated errors will case huge variance in the resulting data. I am afraid this has to be fixed before discussing the science behind Mn incorporation into foraminifera.

When we would do these analyses today we might follow a different analytical approach. The reason for using the NIST610 standard at that time was the large range between species and with depth we had observed already. Moreover, studies into the fundamentals of laser ablation ICP-MS analyses at the ETH had at the time shown that the absorption behaviour of the lower concentrated NIST lead to significant different particle size distributions, which could influence results, especially when analyzing foraminiferal tests.
We agree that for the lower concentrations pressed powders or an alternative matrix-matched standard might have been better. These analyses are, however, already performed some time ago and to our opinion we did all possible effort at that time to make sure we included as much cross-checks as possible. We used for instance a matrix-matched carbonate standard which we also analysed off line to check our results. The analyses have been performed in two independent laboratories (ETH, Switserland and Utrecht University, The Netherlands), with different machines (Micromass Platform and Elan 6100) and using different software packages (Glitter and LamTrace). Results of cross calibrated samples were identical as well as the results of the matrix matched standard. The error is admittedly larger than what it would be when we do these analyses these days, but in view of the large observed differences they still are very useful.
We have now added several lines to the methods section explaining the followed procedure and potential caveats at lines 181-185: 'For Mn this standard showed a precission better than 3% over all analyses, at ETH and UU, and with an offset of less than 5% from an off line determined (solution ICP-AES) concentration analyzing discrete sub-samples. The matrix matched standard is routinely included in the analyses and has been monitored since 2010 (Duenas Bohorquez et al., 2011).'

Minor comments
- Measurements in the lab should be validated by their lab publication/s therefore sections 211-215 cannot assure the accuracy of the measurements. It has been mentioned in lines 181-182 that calcite standard were analysed for consistence. Data for this reproducibility will be the best indication of accuracy and reproducibility of the method and has to be reported.

The publications we refer to in lines 211-215 are actually from our lab. This is now indicated in this section and also a reference to Duenas Bohorquez et al. (2011), which gives data on the long time accuracy of standards. The accuracy for the matrix-matched calcite standard has been added to the manuscript at lines 181-185

- Section 2.5/2.5.1/2.5.2 (lines 217-239). It is necessary to break this down in sections if there are only 2 sentences in each section?

These sections have been combined into a single section 2.5.

[revised manuscript text omitted]